# Techno-Economic Evaluation of Downdraft Fixed Bed Gasification of Almond Shell and Husk as a Process Step in Energy Production for Decentralized Solutions Applied in Biorefinery Systems

Luís Carmo-Calado [1,2], Manuel Jesús Hermoso-Orzáez [3,*], José La Cal-Herrera [4], Paulo Brito [1,2] and Julio Terrados-Cepeda [3]

1   Polytechnic Institute of Portalegre, 7300-110 Portalegre, Portugal; luis.calado@ipportalegre.pt (L.C.-C.); pbrito@ipportalegre.pt (P.B.)
2   VALORIZA, School of Technology and Management, Polytechnic Institute of Portalegre, 7300-555 Portalegre, Portugal
3   Department of Graphic Engineering Design and Projects, Universidad de Jaen, 23071 Jaen, Spain; jcepeda@ujaen.es
4   Department of Business Organization, Marketing and Sociology, Universidad de Jaen, 23071 Jaen, Spain; jalacal@ujaen.es
*   Correspondence: mhorzaez@ujaen.es; Tel.: +34-610-389-020

**Abstract:** The objective of the present study was to carry out a technical study of the gasification of almond shells and husks at different temperatures and, subsequently, an economic analysis for the in situ installation of a decentralized unit to produce electricity, through a syngas generator, that would overcome the use of fossil fuels used in this agroindustry. The gasification tests were carried out at three different temperatures (700, 750 and 800 °C) and the results for the tests carried out were as follows: a 50:50 mixture of almond husks and shells was found to have a lower heating value of value of 6.4 MJ/Nm³, a flow rate of 187.3 Nm³/h, a syngas yield of 1.9 Nm³/kg, cold gas efficiency of 68.9% and carbon conversion efficiency of 70.2%. Based on all the assumptions, a 100 kg/h (100 kWh) installation was proposed, located near the raw material processing industries studied, for an economic analysis. The technical–economic analysis indicated that the project was economically viable, under current market conditions, with a calculated net present value of k€204.3, an internal rate of return of 20.84% and a payback period of 5.7 years. It was concluded that thermal gasification is a perfectly suitable technology for the recovery of raw materials of lignocellulosic origin, presenting very interesting data in terms of economic viability for the fixed bed gasification system.

**Keywords:** gasification; residual biomass; fixed bed; techno-economic analysis

## 1. Introduction

Biomass residues, particularly agricultural residues, are considered some of the most accessible and cost-effective fuels that can be utilized today [1]. The global interest in the large-scale production of clean and sustainable energy from agricultural residues is increasing, driven by growing concerns regarding the environmental impacts of traditional fossil fuel and nuclear usage [2]. In Spain, biomass consumption for energy production is primarily limited to heat applications, with the exception of 4.6% for electricity generation, mostly associated with large factories that use biomass as a raw material, such as the paper industry [3].

The fundamental implication is that biomass resources come from diverse and varied sources—hence their heterogeneity. This means that biomass can be obtained from a wide range of organic materials, such as agricultural residues, forest residues, energy crops, and even urban waste [4]. Today, there is a technology capable of converting biomass

into bioenergy products that can effectively replace any form of conventional energy. This includes the substitution of solid fuels such as coal and firewood, liquid fuels such as gasoline and diesel, and gaseous fuels such as natural gas [5]. While biomass may not exhibit the same level of versatility as fossil fuels in terms of direct applications, its ecological, social, and economic merits, coupled with ongoing technological advancements, hold the potential to substantiate a more sustainable and diversified energy matrix. It is imperative to contemplate these facets when assessing the feasibility of distinct energy sources. This highlights that biomass can play a significant role in providing an energy supply in various sectors, including residential, commercial, and industrial [6]. In addition, biomass combustion generally exhibits carbon neutrality, but the reduction in air pollutants is not as straightforward and can be influenced by various factors. When comparing specific instances, such as a wood stove versus a gas stove, the emission of air pollutants may not consistently favor biomass combustion. The variability in air pollutant emissions can be attributed to factors such as combustion efficiency, the technology utilized, and the specific characteristics of the biomass being burned. In some cases, gas stoves might exhibit lower emissions of certain air pollutants compared to certain biomass-burning appliances [7]. The most significant achievement so far has been attaining the highest ethanol concentration of 13.2 g $L^{-1}$ during co-current continuous syngas fermentation. Despite its promise as a technology, the process has faced challenges during its scaling up for industrial applications [8]. Alternatively, syngas can be converted into methanol, synthetic natural gas, Fischer–Tropsch liquids, or mixed alcohols by using different catalysts [9]. These chemicals can be further processed or purified to obtain marketable products or intermediates for other industries. Therefore, integrating gasification with a biorefinery system can create a synergistic effect that can improve the efficiency, sustainability, and profitability of biomass utilization.

Despite the global energy evolution, there is still a significant portion of humanity that lacks regular access to electricity [10]. A substantial proportion of this population includes people living in decentralized regions, where the demand for national grid electrification does not justify the required investment. According to the International Energy Agency (IEA), approximately half of the electricity access for these communities will need to be provided through off-grid solutions [11]. This situation arises due to the difficulty of extending conventional electrical infrastructure to remote, sparsely populated, or economically disadvantaged areas. The construction and maintenance of transmission lines and substations in these regions often become economically unviable, leading to the need for decentralized alternatives. Another issue is the excess energy produced by these solutions and how to store it [12]. To adopt these decentralized energy solutions, including distributed generation and isolated systems, it is possible to provide electricity in a more efficient, accessible, and sustainable manner for these communities [13]. For example, conventional diesel generators are used to produce electricity in more remote areas; however, this can lead to higher production costs and additional expenses related to their transportation and environment [14].

Renewable energy sources are gaining recognition as an increasingly clean, reliable, and efficient option for decentralized electrification. Unlike traditional energy sources such as diesel generators, renewables do not rely on fossil fuels and have a lower environmental impact. With advancements in this field, a wide range of small-scale off-grid solutions are becoming available, including biomass processes, wind generators, photovoltaic (PV) solar systems, hybrid systems, and even fuel cells [15,16]. Despite the significant technological and performance advantages of wind and solar energy, they become limited due to their high dependence on atmospheric phenomena [17,18]. Biomass, and especially agricultural byproducts, plays a significant role as a renewable energy source for electricity generation, which can be highly competitive if there is availability of locally accessible feedstock. According to the International Renewable Energy Agency (IRENA), it is estimated that, by 2030, biomass will have the most significant role in the global renewable

energy consumption, surpassing wind and photovoltaic solar energy, with an estimated 77.3 EJ/year [19].

Regarding the current state of bioenergy in Spain, the Ministry for the Ecological Transition and the Demographic Challenge (MITECO) has awarded contracts for 146 MW of biomass and 31 MW of photovoltaic solar capacity. The distributed photovoltaic solar projects, with a capacity of up to 5 MW, were awarded with a winning average tariff of €53.88 per MWh, with minimum and maximum tariffs of €44.98 per MWh and €62.5 per MWh, respectively. In the case of biomass, the weighted average tariff was €93.09 per MWh, with minimum and maximum tariffs of €72.38 per MWh and €108.19 per MWh, respectively. MITECO justified the higher biomass tariffs compared to other renewable sources due to its manageable generation capacity and added value in job creation, especially in rural areas, in addition to aiding in the recovery of forest and agricultural residues [20].

For the installation of a gasification unit, factors such as fuel availability and seasonality, maintenance, transportation, infrastructure, and equipment maintenance must be taken into consideration [20,21].

In the mentioned context, almond byproducts, such as almond shells, pruning, and outer almond husks, have high potential for energy recovery in many countries. This is due to their relatively high energy content, ranging between 16 and 19 MJ/kg [22]. These byproducts are considered a valuable source of biomass, which can be utilized for energy production, particularly in situ. In Spain, specifically, there is significant production of these byproducts, reaching approximately 3.4 million tons per year [23]. This number highlights the relevance of almond byproducts as a biomass source for energy production. The efficient utilization of these byproducts by the industry can contribute to the diversification of the energy matrix while promoting clean energy, reducing the dependency on fossil fuels and the costs associated with almond processing.

The gasification of agricultural residues has garnered significant interest as a thermochemical conversion technology when compared to the direct combustion of the residues. This is due to its higher efficiency compared to conventional combustion, which is one of the main disposal methods for agricultural residues [24]. Gasification allows the production of a synthesis gas rich in hydrogen, carbon monoxide, and a small percentage of methane and other light hydrocarbons, which can be directly used to generate energy or to produce other chemicals. The synthesis gas, or syngas, has significant potential as a source of clean and renewable energy [25]. This process occurs in a controlled atmosphere of steam and/or air, where the fuel is subjected to high temperatures in the absence of oxygen, resulting in the thermochemical decomposition of its components. In the specific case where air is present during the gasification process, the ratio of oxidizing agent to biomass ranges between 0.2 and 0.4 [26]. The temperatures of this thermochemical process range from 700 °C to 1500 °C and have a significant influence on the gasification reaction and, consequently, on the composition of the syngas [27]. Temperature is a critical factor for the gasification and combustion of almond shells and husks, as the reaction rate increases as the temperature rises. For this reason, the present study, supported by the literature, validates the conversion characteristics and the energy induced by the reaction affected by a temperature gradient in an air atmosphere and relies on an analysis of almond shell and husk application under pilot-scale conditions in a downdraft fixed bed reactor of approximately 100 kW$_\text{electric}$ [28]. However, it is crucial to emphasize that the gasification of agricultural byproducts—in this case, almond shells and husks—faces not only technical challenges but also economic challenges. Process efficiency, syngas characteristics, appropriate fuel selection, and financial viability are just some of the aspects that must be considered when implementing this technology.

The selection of a small-scale gasification unit (100 kg/h) with a specific focus on utilizing agro-waste presents a multitude of advantages that substantiate its consideration as a propitious solution. The initial advantage stems from the utilization of waste materials, thereby mitigating the necessity for disposal and effectively contributing to the sustainable

management of these two waste streams. After this, the aspect of decentralization and reduced energy dependency comes to the forefront. In this regard, the deployment of small-scale gasification facilities facilitates localized energy generation, thereby diminishing the reliance on extensive central power generation infrastructures and transmission networks. This attribute assumes particular significance in rural or isolated areas characterized by constrained access to energy resources, concurrently fostering diversification within the energy portfolio. Moreover, the advent of small-scale gasification units contributes to the stimulation of indigenous economic prospects. The impact of a unit of this type establishes the creation of businesses linked to the operational and maintenance aspects of these facilities, thus energizing the economic activities in the immediate vicinity and, simultaneously, providing a propitious environment for the promotion of auxiliary businesses and industries synergistically aligned with the gasification domain. Conversely, this equipment presents facile maintenance and modulatory capacity, intrinsic to small-scale gasifiers, engendering a number of benefits. These encompass heightened operational efficacy, curtailed periods of non-operation, augmented safety protocols, and potential economization endeavors.

Considering the energy needs and concentrated crops in the almond production industry, as well as the high production of readily available and untreated byproducts, installing small-scale energy valorization units becomes a viable and renewable solution in terms of reducing energy costs and the carbon footprint. These decentralized energy alternatives have the advantage of lower initial investment, easy scalability, and suitability for employment in rural areas. Thus, the objective of this study was to technically evaluate almond shell and husk gasification at three different temperatures and its economic feasibility in small-scale or pilot-scale gasification facilities, in collaboration with the industry producing this feedstock. An economic assessment of the entire process was conducted, along with a Monte Carlo sensitivity analysis to determine its economic potential. Finally, a brief conclusion is provided regarding the impacts and prospects of implementing a small-scale gasification system.

## 2. Materials and Methods

This section outlines the adopted methodological approach to investigate almond shell and husk gasification in a fixed bed downdraft reactor. The employed methodology comprises a series of fundamental and experimental analyses aimed at comprehending pivotal aspects of the biomass-to-syngas conversion process. The primary methodological steps encompass ultimate analysis, proximate analysis, the determination of high heat value, the description of the fixed bed downdraft gasification equipment, and syngas analysis using gas chromatography. Each of these analyses is indispensable in assessing the feasibility, efficiency, and characteristics of the almond shell and husk gasification process, thereby contributing to a comprehensive understanding of the involved energetic and chemical transformations.

### 2.1. Raw Materials

The raw materials used in the study were the outer husk (Figure 1a) and the inner shell (Figure 1b) of the almond in a 50/50% mass mixture. The particle sizes ranged from 20 to 50 mm in diameter. The almond byproducts were obtained from the almond production industry, and both laboratory analysis and gasification studies were conducted using the material as received.

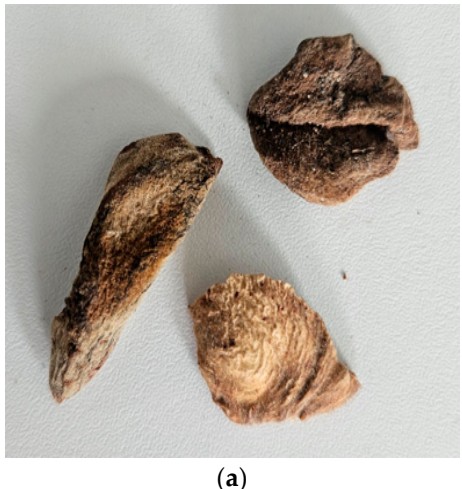
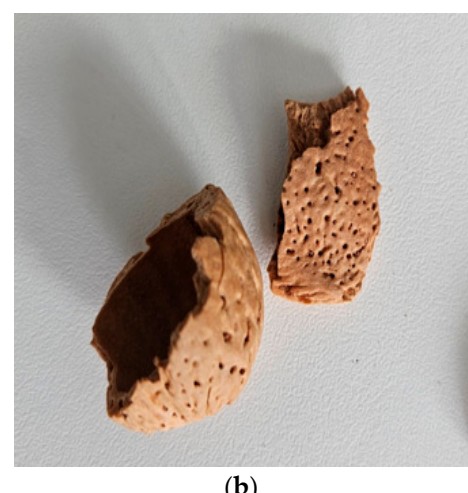

(**a**)　　　　　　　　　　　　　　　　　　(**b**)

**Figure 1.** Almond husk–shell mixture: (**a**) almond husks; (**b**) almond shells.

*2.2. Raw Material and Gasification Product Analysis*

2.2.1. Elemental Analysis (Ultimate Analysis)

Elemental analysis, also known as ultimate analysis, is a process that allows the determination of the basic chemical composition of a raw material. Nitrogen (N), carbon (C), hydrogen (H), sulfur (S), and oxygen (O) content was determined with a Thermo Fisher Scientific Flash 2000 CHNS-O elemental analyzer.

2.2.2. Thermogravimetric Analysis (Proximate Analysis)

Thermogravimetric analysis, or proximate analysis, was performed to determine the concentration of moisture, volatile matter, fixed carbon, and ash (or inorganic components) with a PerkinElmer device, model STA 6000 (PerkinElmer, Madrid, Spain).

2.2.3. Higher Heating Value (HHV)

The higher heating value (HHV) of the byproducts or fuels was determined using the IKA C 2000 calorimetry equipment (Cole-Parmer Instrument, Cambridgeshire, UK).

2.2.4. Gasification Performance

The gasification tests were conducted in a pilot-scale fixed bed downdraft reactor, as illustrated in Figure 2. The gasification system consists of a fuel storage hopper and a heat exchanger. The recirculation of the hot gases generated in the reactor is used to preheat the fuel and remove the moisture present in it. The reactor used in the system is divided into four distinct zones. The first zone is referred to as the drying zone, located at the top of the reactor, where the fuel material is heated to remove the moisture present in it. This is followed by the pyrolysis zone, located approximately in the middle of the reactor, where the fuel material undergoes devolatilization, meaning that the volatile components of the material are released in the form of gases. In the lower part of the reactor, at the reactor throat, responsible for funneling the fuel and introducing the oxidizing agent, combustion reactions occur, providing the thermal energy necessary for the gasification process. The last zone of the reactor is the reduction zone, located at the end of the reactor. Here, the formation of chars takes place, which are solid particles resulting from the partial oxidation of the material. These particles are responsible for the thermal cracking of the gas products, converting them into smaller gaseous products. This division into distinct zones allows each stage of the gasification process to be controlled and optimized to achieve the maximum energy utilization from the fuel used [29]. Attached to the reactor at the bottom, there is a grate-shaped agitator that serves to remove the unconverted material and ashes to a container, with the help of a screw feeder. The gas product is withdrawn from the reactor at a temperature of approximately 500 °C and is cleaned in a cyclone filter. Then,

the syngas passes through the heat exchanger mentioned earlier and is further cleaned in a particle filter of different sizes, where condensates are also retained. After cleaning, the syngas can be directly burned in a flare or introduced into an internal combustion engine.

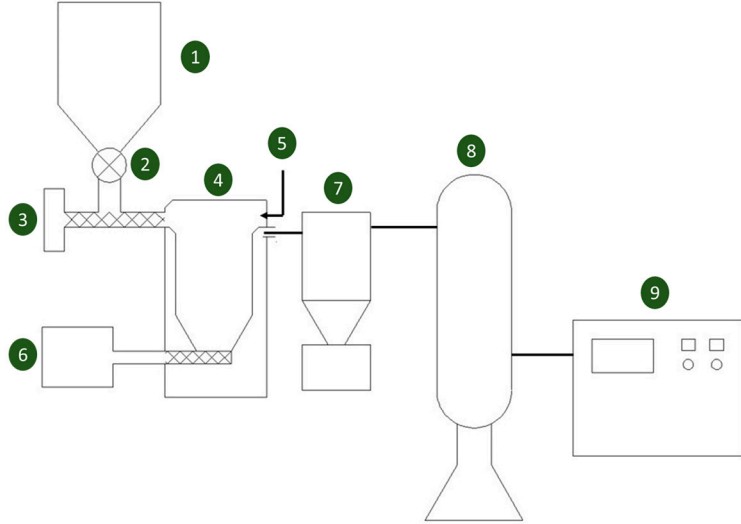

**Figure 2.** Schematic diagram of the experimental apparatus (1—Hopper; 2—Valve; 3—Screw Feeder; 4—Reactor; 5—Air Inlet; 6—Char Container; 7—Cyclone; 8—Particle Filter; 9—Gas Analyzer).

The tests lasted for 420 min, during which the temperature and pressure, fuel and oxidizing agent flow rates, quantity and quality of the produced syngas, and the amount of byproducts (chars and tars) were monitored and controlled.

### 2.2.5. Gas Chromatography—Syngas

During the tests, samples of syngas were collected in Tedlar bags. These samples were subsequently analyzed using a Varian 450GC gas chromatograph. Some of the elements that can be detected and measured by the Varian 450GC include carbon monoxide (CO), oxygen ($O_2$), hydrogen ($H_2$), hydrogen sulfide ($H_2S$), carbon dioxide ($CO_2$), nitrogen ($N_2$), methane ($CH_4$), and other shorter-chain hydrocarbons. This detailed analysis allowed us to obtain accurate information about the composition of the gases present in the samples and was useful for the characterization and evaluation of the quality of the syngas.

### 2.3. Theoretical Parameters

### 2.3.1. Equivalence Ratio

The equivalence ratio (ER) parameter is widely used to quantitatively analyze whether a mixture is rich, lean, or stoichiometric [30]. The ER is a key parameter in the design of a specific gasifier and also significantly influences the quality of the products resulting from gasification. The ER is defined by the ratio between the actual amount of air and fuel used and the stoichiometric ratio of air and fuel [31]. Generally, this term is applied in situations of air deficiency, such as those found in a gasifier, and can be expressed by the following equation (Equation (1)):

$$ER = \frac{(A/F)}{(A/F)stoichiometric} \tag{1}$$

where ER refers to the equivalence ratio; (A/F) is the air–fuel ratio under the experimental conditions adopted; and (A/F)stoichiometric represents the air–fuel ratio under stoichiometric conditions. The equivalence ratio parameter is an indicator of the gasifier's performance, as mentioned. The value of ER typically ranges from 0.2 to 0.4, reflecting the appropriate ratio of air to fuel used in the process. In contrast, in the pyrolysis process, where there is no oxidizing agent present, the value of ER is equal to 0 [32].

### 2.3.2. Cold Gas Efficiency

To assess the efficiency of the gasification process, a commonly used parameter is cold gas efficiency (CGE). CGE is a measure of how efficiently the fuel is converted into syngas during gasification. It indicates the proportion of the fuel that is converted into useful gas [33]. In the study, it is assumed that the calculations of cold gas efficiency (CGE) are based on the lower heating value (LHV), which takes into account the presence of water vapor in the syngas and includes the fuel used in the calculation. Therefore, it was necessary to calculate the LHV of the fuel before calculating the CGE. To calculate the LHV of the fuel, Equation (2) was used. This equation allows the determination of the value of LHV based on the energy characteristics of the fuel used [1,34]:

$$\text{LHV}_{\text{fuel}}\left(\frac{\text{MJ}}{\text{kg}}\right) = \text{HHV} - 0.212 \times \text{H}_2 - 0.0245 \times \text{M}_\text{o} - 0.008 \times \text{O}_2 \tag{2}$$

where $\text{LHV}_{\text{fuel}}$ is the lower heating value of the fuel, HHV is the higher heating value of the fuel obtained from the calorific value analysis, $\text{H}_2$ is the percentage of hydrogen obtained in the CHNS-O analysis, Mo is the percentage of moisture in the fuel obtained with the thermogravimetric balance, and $\text{O}_2$ is a percentage of oxygen obtained in the CHNS-O analysis.

Equation (3) is used to determine the CGE:

$$\text{CGE}(\%) = \frac{\dot{\text{m}}_{\text{syngas}} \times \text{LHV}_{\text{syngas}}}{\dot{\text{m}}_{\text{fuel}} \times \text{LHV}_{\text{fuel}}} \times 100 \tag{3}$$

where $\dot{\text{m}}_{\text{syngas}}$ is the mass flow rate (kg/h) of the syngas and $\dot{\text{m}}_{\text{fuel}}$ is the mass flow rate of the fuel (kg/h).

### 2.3.3. Syngas Yield

The syngas yield ($\eta_{\text{syngas}}$) refers to the volumetric flow rate of syngas produced per unit mass of fuel (Equation (4)) [35]:

$$\eta_{\text{syngas}}\left(\frac{\text{m}^3}{\text{kg}}\right) = \frac{\dot{\text{m}}_{\text{volumétricsyngas}}}{\dot{\text{m}}_{\text{massicfuel}}} \tag{4}$$

where $\dot{\text{m}}_{\text{volumetric}}$ syngas refers to the volume of syngas produced ($\text{Nm}^3/\text{h}$) and $\dot{\text{m}}_{\text{massic}}$ refers to the mass of fuel gasified (kg/h).

### 2.3.4. Carbon Conversion Efficiency

Carbon conversion efficiency (CCE) is a measure that reflects the proportion of carbon contained in the feedstock that has been transformed into syngas. This efficiency can be calculated by considering the volumetric percentages of carbon monoxide, carbon dioxide, methane, and ethylene in the syngas, along with the mass percentage of elemental carbon present in the sample ($\%\text{C}_{\text{sample}}$), expressed in Equation (5).

$$\text{CCE}(\%) = \frac{\eta_{\text{syngas}} \times 12 \times \left[(\%\text{CO} + \%\text{CO}_2 + \%\text{CH}_4) + (2 \times \%\text{C}_2\text{H}_4)\right]}{22.4 \times \%\text{C}_{\text{sample}}} \times 100 \tag{5}$$

### 2.4. Economic Analysis

An economic analysis is established to evaluate the economic viability of the project from an investor's perspective over a predefined lifespan. In order to align this analysis with practical application, this study is built upon a literature review of investment projects in small-scale gasification facilities [36]. Given the significance of the almond industry in the Iberian Peninsula, it is anticipated that a unit of approximately 100 $\text{kW}_{\text{eletric}}$ will be integrated into a larger-scale almond processing industry to mitigate impacts related to transportation costs and supply operation profitability. An existing nearby power line is

also presumed for unit grid connection. A capital cost of €1800/kW is assumed, as it stems from the economy of scale. The plant's operational lifespan is set at 10 years, as significant refurbishment is likely to be required after a decade, according to the manufacturer [37].

The economic assumptions employed to construct the spreadsheet-based economic model were tailored to calculate the net present value (NPV) (Equation (6)), internal rate of return (IRR) (Equation (7)), and payback period (PBP) of the project [38]. The cash flows considered for cost and revenue calculations encompassed the following: initial investment (equity and borrowed capital); operation and maintenance (O & M) costs; and revenue from electricity sales to the grid. All cash flows, except the initial investment occurring solely during the project's startup phase, extend over the 10-year project lifespan.

The net present value (NPV) is calculated to determine the current value of an investment and its profitability. The NPV calculation involves discounting the entire cash flow of an investment to the present value, using a discount rate known as the minimum attractive rate of return (MARR). This process reflects the adjustments needed to bring future cash flows to their equivalent value in the present time.

$$\text{NPV} = \sum_{j=1}^{n} \frac{CF_j}{(1 + \text{MARR})^j} - \text{II} \tag{6}$$

where $FC_j$ is the cash flow, MARR is the minimum attractive rate of return, II is the initial investment, j is the period of each cash flow, and n is the final period of investment.

The internal rate of return (IRR) functions as a discount rate, adjusting values to the initial investment time point. This is distinct from interest rates, where the final value is compounded or accumulated. The computation involves summing each cash flow inflow minus the initial investment, where this summation equals zero.

$$\text{IRR} = \sum_{i=1}^{n} \frac{CF_i}{(1 + \text{MARR})^i} - \text{II} \tag{7}$$

where i is the period of each investment.

The payback period represents an indicator of the time required for an investment to yield returns. It pertains to the duration within which a company will recover the funds invested in a new project or venture back into its coffers.

The aggregate annual cash flow is ascertained by the summation of all incurred costs and generated revenues within each respective year. The annual revenue emerges from the product of the annual electricity output and the prevailing electricity sales tariff during the corresponding year. The annual cash flow is intricately determined by reconciling the aggregate annual costs, revenues, and electrical savings. Subsequently, the cumulative net present value (NPV) is calculated, providing a comprehending assessment of the composite present value of both favorable and adverse cash flows inherent in the investment.

This comprehensive analytical process is systematically executed under the IAPMEI, I.P.—Agency for Competitiveness and Innovation, with considerations covering prevailing price levels, revenues, and prevailing value-added tax rates.

## 3. Results

### 3.1. Almond Husk and Shell Analysis

Table 1 presents the results obtained from elemental analyses, thermogravimetric analyses, and the heating value of the fuel under study.

**Table 1.** Almond husk and shell proprieties.

| | Parameters | Unit | Shell [1] | Husk [1] |
|---|---|---|---|---|
| Ultimate | C | % | 55.2 | 43.1 |
| | H | | 6.4 | 5.7 |
| | N | | 0.2 | 3.3 |
| | S | | 0 | 0 |
| | O | | 35.3 [2] | 36.1 [2] |
| Proximate | Moisture | % | 9.8 | 11.3 |
| | Volatile | | 58.2 | 57.7 |
| | Fixed Carbon | | 29.1 | 19.2 |
| | Ashes | | 2.9 | 11.8 |
| HHV | | MJ/kg | 18.7 | 16.1 |

[1] As received. [2] Calculated by difference.

A comparative analysis of the almond husk and shell in terms of ultimate analysis reveals that the almond shell has a higher proportion of carbon compared to the almond husk. This could indicate that the almond shell contains more carbon-rich compounds, such as lignin, cellulose, and other polymers. Regarding hydrogen, the almond shell has a slightly higher proportion of hydrogen compared to the almond husk. Hydrogen is present in various organic compounds, including carbohydrates and proteins. As for nitrogen content, the almond husk contains a significantly higher amount of nitrogen compared to the almond shell. The presence of nitrogen suggests the existence of nitrogen-containing compounds, such as proteins, amino acids, and possibly other organic compounds. The oxygen concentrations show similar proportions of oxygen in their compositions. The almond shell, with its higher carbon content, might be more suitable for energy generation processes, such as biomass utilization. The substantial presence of nitrogen in the almond husk could make it more suitable for composting or as a nutrient source for soils. The presented data illustrate that both samples exhibit elemental characteristics akin to lignocellulosic biomass [39].

In terms of proximate analysis, the almond husk exhibits slightly higher moisture content (11.3%) in comparison to the almond shell (9.8%). Both the almond shell and husk show similar volatile matter content, with the almond husk displaying a slightly lower value (57.7%) in contrast to the almond shell (58.2%). The almond shell demonstrates significantly greater fixed carbon content (29.1%) when juxtaposed with the almond husk (19.2%). Moreover, the almond husk has substantially higher ash content (11.8%) than the almond shell (2.9%). This higher ash content within the almond husk could potentially impact its suitability for specific applications, as heightened ash content might result in increased residue during thermochemical processes. These observations are depicted in Figure 3. Generally, the almond shell is characterized by higher fixed carbon content and lower ash content in comparison to the almond husk. This suggests that the almond shell could potentially contain higher levels of lignin and cellulose (as illustrated in Figure 4); the degradation of lignin can also be observed in the almond shell sample between 450 and 600 °C and in the almond husk sample between 550 and 700 °C, which is closely related to the fixed carbon present in the sample, thereby presenting greater potential for energy generation processes such as combustion or gasification due to its elevated fixed carbon and reduced ash content. Conversely, the augmented ash content within the almond husk might pose challenges for energy production, due to the increased presence of ash residue.

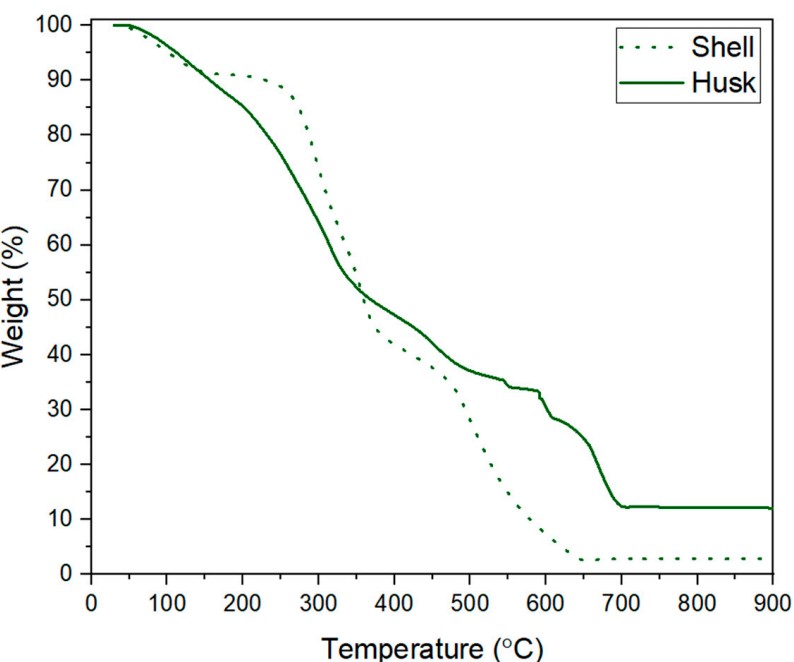

**Figure 3.** TGA curve of almond shell and almond husk.

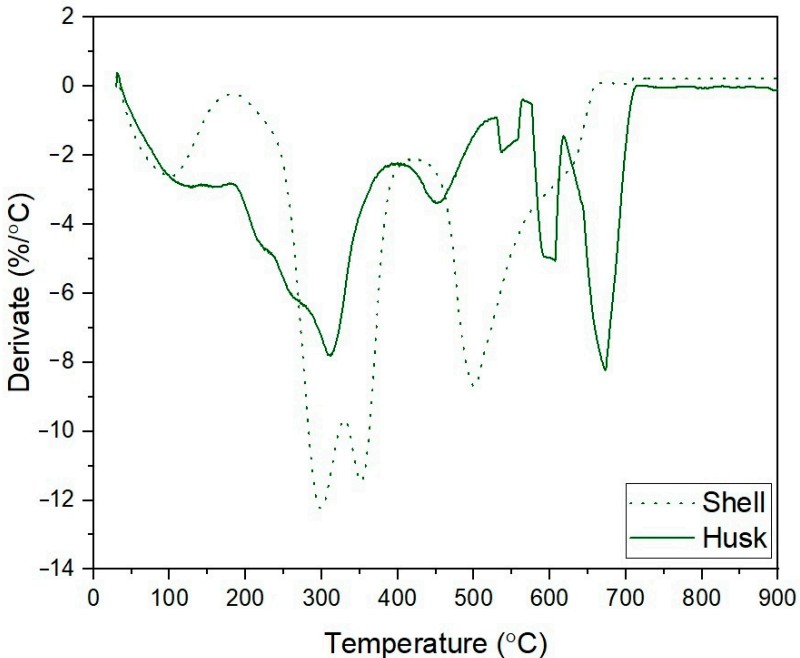

**Figure 4.** DTG curve of almond shell and almond husk.

### 3.2. Gasification Performance and Analysis

Table 2 presents the results obtained from experiments conducted to analyze the efficiency and potential of this technology. It is of note that the data concerning the gas analyses for each sample pertain to the average of the six collected Tedlar bags at each temperature interval. While T1 designates the temperature near the oxidation zone, T2 represents the temperature prevailing within the reduction zone. Additionally, "$P_{react}$" denotes the pressure within the reactor enclosure.

**Table 2.** Almond byproduct gasification results and parameters.

| Parameter | Unit | Temperature | | |
|:---:|:---:|:---:|:---:|:---:|
| $CO_2$ | % | 10.9 | 10.7 | 11.3 |
| $C_2H_4$ | % | 2.5 | 2.5 | 0.8 |
| $C_2H_6$ | % | 0.5 | 0.2 | 0.0 |
| $C_2H_2$ | % | 0.1 | 0.1 | 0.2 |
| $H_2S$ | % | 0.0 | 0.0 | 0.0 |
| $N_2$ | % | 51.7 | 52.1 | 53.2 |
| $CH_4$ | % | 3.6 | 2.9 | 1.1 |
| CO | % | 16.4 | 17.6 | 17.9 |
| $H_2$ | % | 14.4 | 14.5 | 15.5 |
| LHVsyngas | $MJ/Nm^3$ | 6.7 | 6.4 | 4.9 |
| T1 | °C | 748.0 | 794.0 | 851.0 |
| T2 | °C | 632.0 | 615.0 | 629.0 |
| $P_{React}$ | KPa | −32.1 | −30.9 | −39.2 |
| $V_{air}$ | $Nm^3/h$ | 175.7 | 178.3 | 179.8 |
| $T_{air}$ | °C | 21.0 | 22.0 | 20.0 |
| $V_{tar}$ | g/kgfuel | 0.048 | 0.044 | 0.046 |
| $Q_{char}$ | g/kgfuel | 140.0 | 137.0 | 132.0 |
| ER | - | 0.3 | 0.3 | 0.3 |
| $LHV_{fuel}$ | MJ/kg | | 17.4 | |
| $\eta_{syngas}$ | $Nm^3/kg$ | 1.8 | 1.9 | 2.0 |
| $V_{syngas}$ | $Nm^3/h$ | 182.4 | 187.3 | 197.3 |
| CGE | % | 70.2 | 68.9 | 55.6 |
| CCE | % | 66.5 | 70.2 | 68.2 |
| $Q_{comb}$ | kg/h | 100.0 | 100.0 | 100.0 |
| Residence time | h | | 7 | |

The temperature parameter plays a pivotal role in shaping the composition and energy content of syngas. Upon closer examination, it becomes apparent that temperature adjustments during the gasification process impact the resulting syngas quality, which can be observed in Figure 5. As the temperature increases, a discernible shift occurs in the concentrations of its constituents. Specifically, hydrocarbons experience a reduction in concentration, while the concentrations of carbon monoxide and hydrogen increase. This phenomenon can be attributed to the varying thermodynamic properties of the compounds involved and thermal cracking. Hydrocarbons possess a higher enthalpy of formation compared to hydrogen and carbon monoxide. This alteration in composition has cascading effects on vital parameters associated with syngas quality. The lower heating value (LHV) diminishes due to the reduced hydrocarbon presence. Correspondingly, the cold gas efficiency (CGE), a parameter indicative of the effectiveness of the gasification process, experiences a decline as well. This reduction in CGE is directly tied to the decreasing LHV, thereby underscoring the interplay between temperature, composition, and energy metrics. However, the influence of temperature extends beyond composition and energy content. The yield of syngas, a key measure of the quantity produced during gasification, exhibits augmentation with increasing temperature. This upsurge in syngas yield is accompanied by a decrease in the formation of tars and chars, which are undesired byproducts that can impede downstream processes and degrade equipment.

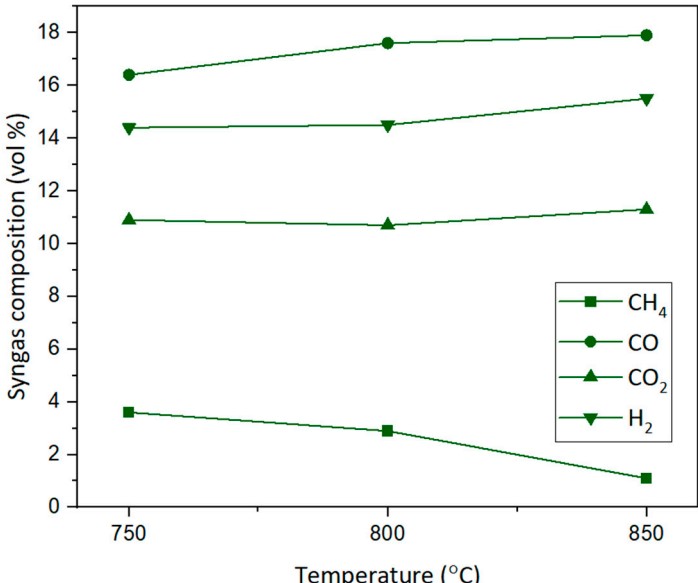

**Figure 5.** Temperature effect on syngas components.

## 4. Discussion

### 4.1. Effect of Temperature on Gasification Tests

Temperature is an important and fundamental variable in various aspects of the gasification reaction, including the syngas yield, carbon conversion efficiency (CCE), heating value of the produced gas, cold gas efficiency (CGE), and quantity of byproducts generated. The composition of the synthesis gas is also strongly influenced by temperature. We can observe the concentration of the main syngas constituents relative to temperature (Figure 5) and the gas yield, CCE, heating value, and overall process efficiency (CGE) in relation to temperature.

As observed, the concentration of carbon monoxide (CO) increases as the temperature rises, due to the enhancement of the Boudouard reaction [40]. The promotion of the Boudouard reaction should have the opposite effect on the concentration of $CO_2$, which should decrease with increasing temperature. However, to maintain a high temperature in the gasification process, an increase in the oxidizing agent is required, resulting in an increase in $N_2$ and $CO_2$, the latter through the oxidation reaction (Equation (8)). This aspect may be related to the reaction in which the generated $CO_2$ reacts with carbon or char to produce 2CO, as observed in Equation (9).

$$C + CO_2 \leftrightarrow 2CO \tag{8}$$

$$C + O_2 \leftrightarrow CO_2 \tag{9}$$

Regarding the concentrations of $H_2$ and $CH_4$, distinct behavior is observed, with $H_2$ tending to increase as the temperature rises, while $CH_4$ decreases. This increase in $H_2$ can be attributed to the enhancement of thermal cracking reactions as the temperature rises, which reduces the amount of light hydrocarbons undergoing this cracking process. On the other hand, the decrease in $CH_4$ can be explained by Le Chatelier's principle. According to Le Chatelier's principle, when a disturbance is applied to a system at equilibrium, the system responds in such a way as to counterbalance this disturbance and restore equilibrium. In this context, the increase in temperature can shift the equilibrium of the reactions, favoring the direction of $CH_4$ towards additional reactions that convert it into thermal cracking products, such as $H_2$. These opposite trends in the concentrations of $H_2$ and $CH_4$ reveal the complexity of the reactions involved in the gasification process and the importance of temperature as a decisive variable that influences reaction rates and equilibria [41,42].

Regarding the performance parameters of the gasification process, Figure 6 shows the evolution of the carbon conversion efficiency (CCE), lower heating value (LHV) of syngas, syngas yield, and cold gas efficiency (CGE) with respect to the gasification temperature. The LHV of the syngas decreases with the increasing temperature, mainly due to the maximum yield of hydrocarbons formed, which have a higher formation enthalpy and are more present at lower temperatures [43].

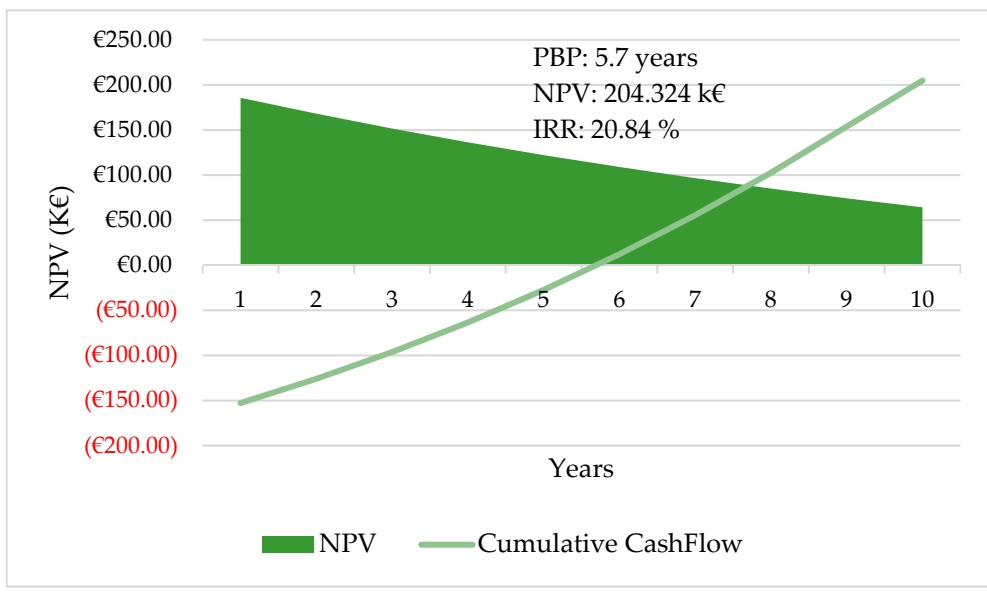

**Figure 6.** NPV profile variation, IRR and PBP for the gasification plant.

As for the CGE parameter, the best performance occurred at 750 °C, with CGE decreasing as the temperature increased. The obtained results are directly related to the characteristics of the fuels used in the gasification process, as well as the importance of the oxidizing agent. The thermal cracking reactions, along with the Boudouard reactions, are the main factors responsible for the observed outcomes. Consequently, the increase in the amount of oxidizing agent had a significant impact on the final volume of the produced synthesis gas, as well as on the LHV of the syngas, resulting in a decrease in CGE as the temperature rose [44]. The experimental results indicated an increase in the volume of the produced synthesis gas as the temperature increased, for all tested conditions. In other words, the experiments showed that by modifying some key variables in the gasification process, such as the amount of oxidizing agent and consequently the temperature, there was a significant decrease in the overall process efficiency (CGE). The opposite is observed for the syngas, which is dependent on the volume of syngas and not its energy content. For temperatures of 750 °C and 800 °C, it will be necessary to consume more fuel to produce the same volume of syngas as at 850 °C. This observation may be related to the increase in oxidizing agent to maintain the temperature stabilized around 850 °C [45]. The carbon conversion efficiency (CCE) reflects the fraction of the sample particles that were transformed into gas during the gasification process. The highest percentage of carbon conversion in the tests was at a temperature of 800 °C (70.2%), whereas the lowest conversion occurred at a temperature of 750 °C. An increase in this parameter may also be intrinsically linked to a reduction in tar formation, as demonstrated in the conducted tests [46].

Based on the study's assumptions, and to gain a more comprehensive understanding of the process, it is imperative to recognize that electrical energy will be generated from the chemical energy (syngas) derived from gasification, with the aim of producing 100 kW$_{electric}$. Operating on the premise that the combined efficiency of the internal combustion engine

and generator is 37% ([47], and in calculating the electrical power, we proceed with the following expression (Equation (10)):

$$P_{electric} = V_{syngas} \times LHV_{syngas} \times \eta_{engine+generator} \times 3.6 \tag{10}$$

where $P_{electric}$ represents the acquired electrical potency, $\eta_{engine+generator}$ symbolizes the efficiency of the internal combustion engine fueled by the syngas and interconnected generator, and 3.6 stands as the conversion factor for kilowatt-hours (kWh).

The adoption of the 800 °C test as the paradigmatic choice to conduct the economic inquiry finds its rationale in the syngas demonstrating a heightened calorific potential when juxtaposed with the 850 °C test. This preference is additionally substantiated by the attenuated generation of light and elongated hydrocarbon chains, which are the critical instigators of operational problems and detrimental impacts on equipment, in contrast to the 750 °C test. The resultant yield from this test culminates in an approximate 123 kW of electrical capacity, within which a prudent allowance of 10% for inefficiencies is admissible.

The utilization of residual heat from the process presents an avenue by which to harness energy and optimize resource usage within the almond processing sector. However, within the scope of this study, the inclusion of residual heat as a value-added component was not factored in. Moreover, chars exhibit potential utility, particularly in soil remediation initiatives and enhancing water retention in the context of intensive almond cultivation practices. By exploring its application in these domains, coal can serve as an agent of sustainable resource management, fostering ecological equilibrium and supporting the agricultural sector [48].

### *4.2. Economic Analysis*
### 4.2.1. Economic Assessment

An economic analysis was conducted to assess the economic feasibility of implementing a small-scale gasification unit alongside the almond exploration industry from the investor's perspective, considering a predefined lifetime period. Based on the technical data obtained, the analysis assumes a 100 kW$_{electric}$ unit with a capital cost of €1800/kW$_{electric}$ [49,50]. The study considers that the project will have a duration of 10 years, derived from the equipment's useful life, with an operations phase between 2024 and 2033 [51]. Due to the utilization of byproducts from the almond processing industry without the need for pre-treatment and the unit being installed at the processing facility itself, the costs related to the transportation and pre-treatment of the raw material are negligible. For the envisaged system, an estimated annual input of approximately 720 tons will be requisite to yield 100 kW$_{electric}$, thereby engendering an annual electricity output of 720 MWh over a continuous operational duration of 7200 h [36]. It is noteworthy to underscore that within the context of almond cultivation, the yield of almonds within their husks amounts to approximately 2 tons per hectare. Under the presumption that the husk component approximates the total mass, this consequently renders a yield of approximately 1 ton of husks per hectare [52]. Due to the seasonality of the almond harvest, the proper storage of almond shells is of paramount significance to preserve their quality and potential utility. Ideally, almond shells should be stored in a well-ventilated, cool, and dry environment, thereby minimizing exposure to moisture. Storage within a covered facility, shielded from rainfall and ground moisture, is highly recommended. In essence, a dedicated storage facility accommodating substantial quantities of these biomass resources would be imperative for subsequent utilization [53]. For the present study, it is noteworthy that this storage facility has already been integrated within the processing industry framework.

Maintenance is assumed to correspond to 10% per year of the total capital cost of the unit, and operation will require one dedicated employee with a salary corresponding to 10% of the capital cost. However, dependence on the conventional electricity grid makes companies susceptible to fluctuations in electricity prices. Through the commitment to self-consumption, both the industrial entity and the gasification installation create a

stable energy reservoir, thus reducing vulnerability to vicissitudes in the energy sphere. Approximately 20% of the generated electricity will be used for self-consumption, and the remaining portion will be sold to the national electrical grid to make the process profitable. In general, the integration of self-consumption, particularly through technologies such as gasification, can exert a markedly favorable influence on the economic, environmental, and sustainability facets of an industrial entity or unit. This outcome not only produces a reduction in operational expenditures but also reinforces energy resilience, augments the corporate reputation, and contributes to a more sustainable trajectory.

Table 3 details the economic assumptions used to create a spreadsheet-based economic model. This economic model aims to calculate three important investment decision indicators: the net present value (NPV), internal rate of return (IRR), and payback period (PBP). NPV is a metric that seeks to determine the feasibility of an investment project and calculates the difference between the project's future cash flows and the initial investment, bringing these cash flows to the present value based on an appropriate discount rate. IRR is a metric that allows the evaluation of investment projects and represents the discount rate at which the project's NPV becomes zero, i.e., when future gains equal the initial investment. PBP indicates the time needed to recover the initial investment based on the project's cash flows. The shorter the PBP, the faster the investment will be recovered, which is generally considered more favorable. These three indicators are crucial in assessing whether an investment project is viable and potentially profitable. By considering the economic assumptions detailed in Table 3, it is possible to make more accurate financial projections and make informed decisions about the investment.

**Table 3.** Economic assumptions for a 100 kW$_{electric}$ small-scale gasification plant.

| Economic Parameter | 100 kW Unit | Observations |
|---|---|---|
| Inflation rate (%) | 6.8 | Inflation rate applied in May 2023 |
| Initial investment (k€) | 180 | 1800 €/kWh |
| Maintenance and operation cost (k€) | 36 | 20% capital cost |
| Electric energy produced (MWh/year) | 720 | Operation time 7200 h/year |
| Electricity energy sold to the grid (MWh/year) | 576 | |
| Electricity energy sales tariff (€/MWh) | 93.1 | https://energia.gob.es/es-es/Paginas/index.aspx (accessed on 23 August 2020) |
| Self-consumption (MWh/year) | 144 | |
| Electric kW price (€/kWh) | 0.13 | https://endesaopenempresas.com/ (accessed on 23 August 2020) |
| Energy sales annual revenue (k€/year) | 53.6 | |
| Self-consumption annual revenue (k€/year) | 18.7 | |

The analysis is conducted using present values, meaning that prices, revenues, and value-added tax rates are considered in the current context. After the year 2023, the inflation rate applied is determined by the average of the last 10 years. All interest rates considered are based on quotes provided by Portuguese banks for projects similar to this one.

In Figure 6, the results of the economic study are shown, where the technical outcomes resulting from the almond byproduct gasification at a temperature of 800 °C are considered. Nonetheless, all the gasification trials of the almond byproducts revealed remarkable technical characteristics and performance, enabling the performance of this economic study. Through this visual representation, the results of the employed economic model in calculating the net present value (NPV), internal rate of return (IRR), and payback period (PBP) are presented.

Given the current economic scenario, the 100 kW$_{electric}$ gasification project appears as an alternative to current energy sources used in rural areas. The financial analysis reveals results of a singular magnitude, with an estimated NPV value of 204.3 k€, an established IRR of 20.84%, and a PBP duration of 5.7 years. This period represents the time lapse required to recoup the initial investment, providing the project with considerable agility.

Several authors have carried out investigations regarding the economic viability of gasification. For example, Lo et al. carried out an economic analysis of a gasifier powered by palm biomass, palm kernel bark, and mesocarp fiber, capable of producing 3000 kWh per month. Although the operating costs of the examined gasification unit were almost twenty times lower than those of the present study, these researchers yielded an average NPV of 20,000 euros and a PBP ranging from 2 to 3 years. These values come from the carefully minimized operating expenses practiced in the Malaysian context [54]. In contrast, Cardoso et al. examined the feasibility of a 100 kWh gasifier for energy generation from forest residues. Predominantly due to the expenses associated with the costs related to biomass and the costs of the gasification unit, this investigation revealed a negative NPV of 32 thousand euros. However, this same study revealed a 1000 kWh gasifier using the same raw material, but producing a positive NPV of 486 thousand euros, an IRR of 17.44 percent, and a PBP of 7.4 years. These values are potentially influenced by the relatively high value of energy sales, of 121.34 euros per megawatt-hour (€/MWh), when compared to the current study (93.1 euros) [49]. Colantoni et al. also focused on biomass gasification, eliciting NPVs of 33.90 thousand euros for a combined heat and power (CHP) gasification project of 13.6 kWe, 537.07 thousand euros for a 136 kWe installation, and a substantial 13.268, 96 thousand euros for a 1.94 MWe system. Correspondingly, the IRR rates were reported as 10%, 25%, and 71% for the mentioned capacities, respectively. Noteworthy are the high unit costs considered in this study, which in turn accommodate heat recovery and supplementary revenue streams not accounted for in the present investigation [55].

Given the data, it can be concluded that the indicators clearly show that the current project of the gasification unit using byproducts derived from the almond processing industry is economically viable. It shows a positive NPV, a high IRR, and a PBP shorter than the equipment's useful life. The current project is now evaluated beyond the presented financial indicators, seeking attractiveness from the investor's perspective. According to typical financial benchmarks for biomass projects found in the literature, it is desirable for the net present value (NPV) to be positive, the internal rate of return (IRR) to exceed 10%, and the payback period (PBP) to be less than 10 years [56]. Although these criteria may vary depending on the country's risk and project specifics, they will be used as a reference for the analysis. Taking into account the described information, it can be stated that the project has financial robustness and aligns with best practices for investing in biomass projects.

### 4.2.2. Sensitivity Analysis

Sensitivity analysis was performed on five pivotal economic variables, encompassing maintenance, operational expenditures, the initial investment cost, the electricity tariff, and self-consumption. This evaluation is graphically represented through a spider diagram (Figure 7), elucidating the influence of alterations in each key economic factor on the overall project economics. This diagram serves as an instrument in conducting sensitivity analysis due to its capacity to visually describe the magnitude of perturbations in these economic parameters, directly impacting the net present value (NPV) of the project.

The selection of NPV for this analysis stems from its robust applicability in evaluating project feasibility or dismissal. Unlike the internal rate of return (IRR), NPV employs more pragmatically grounded assumptions regarding reinvestment rates [57]. Particularly, among these examined parameters, the electricity tariff exerted the most substantial influence on project economics, thereby occupying the paramount position in the diagram. Immediate in sequence was self-consumption, demonstrating pronounced significance in the project's underlying dependency structure. Conversely, the investment cost, operation, and maintenance overheads were identified as deleterious factors, inducing negative impacts on the project's NPV.

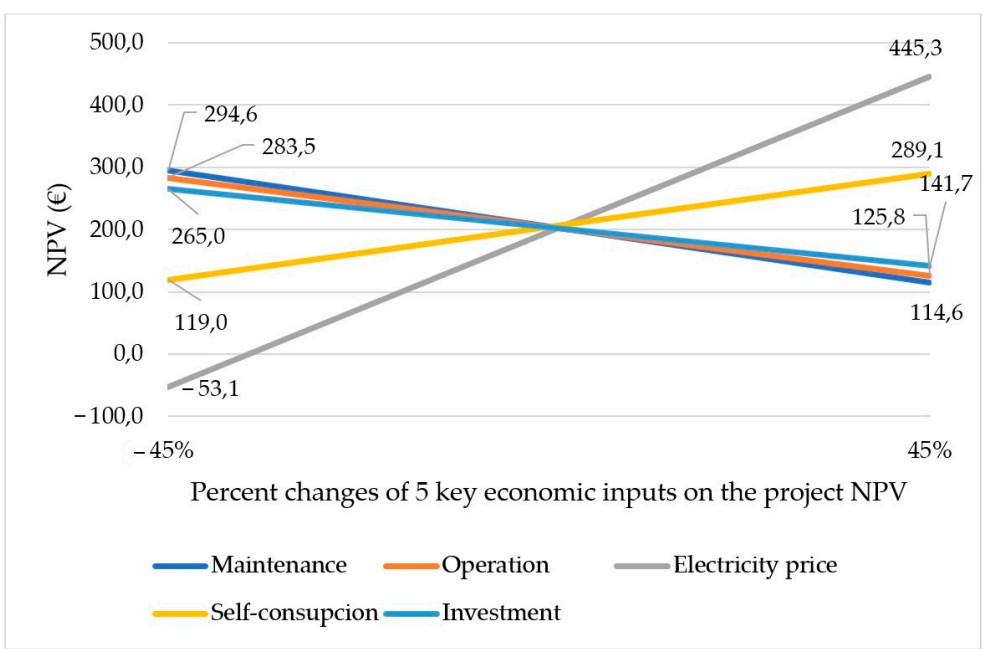

**Figure 7.** Major factors impacting project's NPV.

The operational cost demonstrates a subtle yet discernible negative slope in the sensitivity analysis. Notably, a 45% escalation in labor expenses translates to a significant 38% decrement in the NPV, corresponding to €78k. This underscores the necessity of addressing the impact of labor costs on the project's profitability. One possible avenue for mitigation involves exploring the greater automation of the equipment or leveraging labor resources from the almond processing industry itself, potentially offsetting these negative effects.

In the context of maintenance, the sensitivity analysis reflects a similar negative slope in relation to the project's NPV. A 45% elevation in maintenance costs is observed to correspondingly yield a substantial 43% fluctuation in the NPV, equivalent to €89k. This result underscores the critical importance of effectively managing maintenance expenses to prevent undue strain on the project's financial health.

Contrastingly, the influence of the investment cost, as explored through sensitivity analysis, exhibits a more moderate inclination, impacting the NPV by approximately 30%. This finding suggests that the investment cost holds a comparatively steadier influence on the project's economic outcomes, likely stemming from its role as an upfront expenditure with a less immediate impact on operational performance.

The sensitivity analysis underscores the pivotal role played by the selling price of electricity in shaping the project's economics. The project's pronounced dependency on energy sales is evident, as a 45% increase or decrease in the electricity tariff results in a staggering 125% shift in the NPV, equivalent to €255k. Such significant fluctuations render the project's feasibility highly contingent on maintaining a stable and favorable energy pricing environment.

In a similar vein, self-consumption demonstrates a noteworthy impact on NPV, albeit to a somewhat lesser extent. The sensitivity analysis reveals that self-consumption dynamics, with a 60% influence on NPV, contribute to the overall project economics. This observation underscores the project's reliance on efficient self-consumption mechanisms to optimize its financial performance.

## 5. Conclusions

In the realm of sustainability, the pivotal role of technological advancement in promoting sustainable development and resolving inherent conflicts is undeniable. Challenges stemming from the mismanagement of agricultural and forest lands, coupled with escalating agro-industrial waste, counteract sustainability objectives. Nonetheless, these

challenges can be effectively addressed through the strategic application of technological innovations, management frameworks, and educational initiatives. The emergence of thermal gasification as a solution aligns seamlessly with contemporary industrial needs, addressing environmental concerns and citizen well-being by furnishing a technological avenue for both waste disposal and energy utilization. This innovative approach contributes significantly to diversifying the energy matrix.

The primary objective of this study was to conduct a rigorous technical analysis of almond shell and husk gasification at different temperatures. Subsequently, an economic evaluation was undertaken to assess the viability of an on-site decentralized installation, aimed at reducing the reliance on fossil fuels within the agro-industrial context.

While the preceding sections outlined the technical findings, the focus of this conclusion is to present overarching insights that extend beyond a mere restatement of the results. It is evident that all experiments yielded promising outcomes, warranting consideration for the economic assessment. The analysis led to the selection of a gasification temperature of 800 °C as the preferred option due to its advantageous characteristics encompassing LHV, minimal hydrocarbon presence, and an optimal syngas flow rate. This judicious choice aligns with the objective of maximizing performance while mitigating potential equipment malfunctions.

Based on the comprehensive technical foundation, an on-site 100 kWh installation was proposed for detailed economic scrutiny. Through technical and economic analyses, the project's financial viability was substantiated within prevailing market conditions. Particularly, an NPV of 254.6 k€, an IRR of 22.69%, and a PBP duration of 5.51 years were established, signifying its economic feasibility.

The sensitivity analysis further enhances our understanding of the project's resilience to economic fluctuations. This comprehensive examination highlights the need for multi-faceted cost management approaches, suggesting avenues for automation, efficient maintenance, and strategic resource allocation. The profound impact of the electricity tariff on the project's economics underscores the significance of a stable market environment and diversified revenue streams.

Regarding broader implications, thermal gasification emerges as a potent technology for the valorization of agro-industrial raw materials, exhibiting both economic promise and environmental benefits. While emphasizing the project's attractiveness to potential investors, the study advocates for the deployment of gasification systems as decentralized solutions. These solutions hold the potential to not only diversify the energy matrix but also empower remote populations, driving local economies and sustainable development. Moreover, the adaptability of gasification processes extends beyond electricity generation, presenting opportunities for value-added chemical production.

The study underscores that thermal gasification holds substantial promise for agro-industrial waste valorization, affirming its economic viability. The versatile potential of gasification technology, whether for electricity generation or chemical production, positions it as a catalyst for sustainable progress. The ongoing advancements in this domain, exemplified by projects like HyFuelUP, signify a dynamic future characterized by continued innovation and the pragmatic utilization of thermochemical technologies.

**Author Contributions:** Conceptualization, L.C.-C., J.L.C.-H., P.B., J.T.-C. and M.J.H.-O.; methodology, L.C.-C.; software, L.C.-C.; validation, L.C.-C., J.L.C.-H., P.B., J.T.-C. and M.J.H.-O.; formal analysis, L.C.-C., J.L.C.-H., P.B., J.T.-C. and M.J.H.-O.; investigation, L.C.-C., J.L.C.-H., P.B., J.T.-C. and M.J.H.-O.; resources, L.C.-C.; data curation, L.C.-C.; writing—original draft preparation, L.C.-C.; writing—review and editing, M.J.H.-O.; visualization, M.J.H.-O.; supervision, J.L.C.-H., P.B., J.T.-C. and M.J.H.-O.; project administration, M.J.H.-O.; funding acquisition, J.L.C.-H., P.B., J.T.-C. and M.J.H.-O. All authors have read and agreed to the published version of the manuscript.

**Funding:** This research received no external funding.

**Data Availability Statement:** All data is included in the article.

**Conflicts of Interest:** The authors declare no conflict of interest.

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
