# Peer review of "Techno-Economic Evaluation of Downdraft Fixed Bed Gasification of Almond Shell and Husk as a Process Step in Energy Production for Decentralized Solutions Applied in Biorefinery Systems"

_agronomy, doi:10.3390/agronomy13092278_

Round 1

Reviewer 1 Report

The paper presents the technical results of almond husk and shell gasification, and amends that with a brief assessment of economic feasibility.

In my opinion, the topic of the paper is of high interest, and residues from almond production seem to be a very suitable substrate for gasification due to their high and localized availability and their low moisture content. The gasification procedure and the measurements are generally described well in the study, and the measurement results are relatively clearly presented in two tables.

The manuscript, in its current form, however, could be significantly improved, and I would encourage the authors to do so, to make the research more accessible to more readers.

The main issues are:

1.       The manuscript is unnecessary lengthy in some parts, while it lacks detail in others, where more information is needed.

2.       The text is not split correctly according to the main sections (Introduction, Methods, Results, Discussion). Several methods and results are for instance found in the discussion section.

3.       The economic analysis is not described well, and not described at all in the methods section.

4.       Assumed investment costs are determining the result of the economic assessment, but probably based on outdated information.

Detailed comments:

Line 20-22: “the results for the tests carried out were, for a consumption of 100 kg/h, a lower heating value of 6.4 MJ/Nm3 …” This sounds like these were the conditions for the measurement, while this is actually part of the results already. Would suggest to split this and write something like: “A 50:50 mixture of almond husks and shells was found to have a lower heating value of …”

Line 22: “Based on all the assumptions, a 100 kg/h (150 kWh) installation was proposed, …” Even after reading the whole article I did not understand why these parameters were chosen.

Line 25: “net present value of k€254,6, an …” Check comma.

Abstract in general: Information is missing on what the assumed end product of gasification is: The syngas that would be sold, biochar or electricity?

Line 47: “One of the major benefits of biomass is its versatility in terms of application, as it can be used in combustion facilities of all sizes, …” This is no advantage of biomass compared to fossil fuels (especially oil & gas). They are actually much easier to scale and use for different purposes, such as in a gas cooker or huge gas power plants.

Line 52: “biomass also offers environmental benefits, as its combustion generally leads to lower emissions of greenhouse gases and air pollutants, …“ Please be more precise. Combusting residual and annually produced biomass is generally carbon neutral. However, it does not necessarily lead to lower air pollutants (e.g. wood stove vs gas stove).

Line 80: “For this reason, the study validates …” Which study are you referring to? Yours?

Line 89: I find this switch from the specific focus on almond shells and gasification (lines 65-88) to this very general section on electricity access very strange. In my opinion, the introduction should start general, and then become more specific. This jump between specific and general is not really nice to read. I would suggest to delete the paragraph (lines 89-103) and move the following paragraphs (from 104 onwards) more to the beginning of the introduction.

Line 99: “By adopting such off-grid solutions …” Which off-grid solutions? And how does this fit together: Here you are talking about “off-grid solutions”, later (line 122) you refer to feed-in tarrifs, which obviously require connection to the grid.

Line 145: The method section should start with an overview over your approach. Currently one can only guess what you may have done at this point.

Line 154: This is an example where the text is much too long without providing any meaningful info. I would delete the subheadings 2.2.1 and 2.2.2 and just mention it under 2.2. Instead of the whole paragraph 2.2.1 you could just write something like “Nitrogen (N), carbon (C), hydrogen (H), sulfur (S), and oxygen (O) contents were determined with a Thermo Fisher Scientific Flash 2000 CHNS-O elemental analyser.” I think it is important to mention what a CHNS-O is, but really no more information is needed.

Line 174: “byproducts or fuels” What you you mean here? The husk shell mixture?

Line 219: “This equipment is capable of identifying and quantifying different gaseous elements present in the samples.” I think you shouldn’t assume that the readers are stupid. They will probably know that a gas chromatograph is capable of measuring gases. Just delete this sentence.

Line 274: I am really missing the description of the methods for the economic analysis here.

Lines 276-284: This should not be part of the results section. The results section should really only show results. Please delete this, and instead pick out and describe noteworthy results. I for instance find the low moisture content of husks and shells, and the low ash content of the shells interesting.

Table 1: These results of the ultimate analysis are related to the fresh matter content right? This should be mentioned in the caption. “Uni.” Is this an abbreviation for “Unit” ?? If so, please don’t do that.

Lines 288-289: “gasification, it is an innovative and promising process “ This is the results section!! Please don’t discuss here. Move to discussion or delete.

Table 2: Again “Uni.”. What do the “Sample” columns stand for? The different target temperatures? Then please rename. In line 301 you write of 6 samples, where are these represented in the table? Are the three just a random selection from these six? What is the difference between T1 and T2??

Lines 295-306: This is all methods, and should not appear in the results section. Instead, a textual description of the results is missing.

Line 308: “Theoretical Parameters” Why are these theoretical parameters? Are they not real?

Sections 4.1.1.and 4.1.2: I think these are the least interesting sections. I would shorten them significantly. Also again, please do not assume that the readers are stupid. E.g. “Carbon is the primary constituent of organic materials and plays a fundamental role in the heating value of the fuel.” There is really no need to mention such basic things.

Figures 4, 5, 6: These are clearly results, and should therefore be moved to the results section, while lines 359-378 where you speculate on the process should remain in discussions.

Figure 6: I think having three different scales / units in one graph is not good style.

Lines 387-408: This leaves me confused. Which of the two parameters CCE or CGE is now more important to assess the performance? “The CCE (Carbon Conversion Efficiency) is a metric of utmost relevance …”?? Do you want to say that the CCE is the one that we should pay more attention to?

I was wondering whether there is any difference between the different temperatures in terms of energy input. Do you need to preheat the gasifier somehow? Is there a difference in the energy input required? Are the results presented here for the whole runtime or only for after it has reached the target temperature? How would this affect results, if the gasifier was run 24/7 or for instance started each day in the morning?

Line 414: “4.3. Tcno-economic Analysis” Tcno??? Was Techno to long? Actually, what is coming is only the economic analysis anyhow.

Section 4.2.1: As mentioned before, the economic assessment urgently needs its own section under “Methods”. The description here is too short and lacks important information.

Line 419: Here you mention a capacity of 150 kWe. I assume that the small ‘e’ stands for electric. You have not mentioned before how you would get electricity from the syngas. What were your assumtions for this? Which conversion efficiency did you assume? Did you conduct tests to figure this out? Would there maybe be any use for the residual heat from the syngas combustion? Also, you seem to only consider the use of the syngas, but are there any meaningful quantities of char that could be used? This should be mentioned in the results section as well.

Line 425: “approximately 720 tons per year will be required to obtain 150 kWe …” Is this a realistic size for a plant? For instance how many tons of husks and shells are collected per ha of almond farm, and what is the average farm size? Would this 150 kW plant fit for one farm, or for several together??

Line 426: If you assume 7200 hours that means the plant would be running for the whole year, while the harvest happens in a short period of time. What are your assumptions on storing the husks and shells? Would they rot if they are just piled up? What is the current usage of these residues?

Table 3: For the capital costs you assume 1300 € / kWh. Where is this data coming from? Are you using the regression from Bridgwater et al. 2002? As far as I can see this study only included gasifiers bigger then 1 MW, and costing more than 1 million €. I think you cannot just use this data. Also, the reference is more than 20 years old, and the values probably very outdated. Just from a brief search I could find several other potential sources, which partly use significantly higher values:

Colantoni et al. 2021 (https://doi.org/10.1016/j.egyr.2021.03.028) , referencing Sentis et al. 2016, consider total investment costs of 300.000 € for a 100 kWth gasifier.

Safarian et al. 2020 (https://doi.org/10.4236/jpee.2020.86001) considered 105.500 €, just for the hardware plus additional costs for installation.

Codoso et al. 2020, which are also cited by the authors, assume capital costs of 1760 € /kW for a 100 kW plant.

Allesina and Pedrazzi 2021 (https://doi.org/10.3390/en14206711) Provide an overview on companies and elaborate on maximal feasible  investment costs.

Since you define maintenance and operation costs relative to the capital costs, these assumptions on investment costs determine the whole result of the economic assessment. In such conditions, I would suggest that you do a sensitivity analysis based on a thorough literature research on potential capital costs.

Table 3 also mentions an “Electric kW price (€/kWh)”. What is this used for? Is it assumed that the plant consumes electricity from the grid? As far as I can see this is not mentioned anywhere.

Line 453: You write that the results are for 800°C. Why was this temperature taken? Is this the best temperature based on the previous results? If so, I would like to understand why.

Figure 7: You are using inconsistent abbreviations of Payback Period in the text and figure. Sometimes PB, sometimes PBP.

Lines 461-464: “Thus, Figure 7 personifies the joint effort between technical robustness and economic excellence, shedding light on the path for informed and well-founded decision-making. In this scenario of innovation and relentless pursuit of sustainable ventures …” This is not appropriate language for a scientific publication.

The discussion lacks a comparison to the multitude of other studies on biomass gasification. There are dozens of other studies that have addressed similar questions. E.g. Colantoni et al. 2021 have considered hazelnut shells. Do they come to similar results and conclusions? If not, why?

I think the conclusions section should really focus on the conclusions, and not reiterate the results.

Lines 541-554: The final document should probably not contain the instructions for filling out this section.

I think the use of English is generally ok. In some cases, which I have pointed out above (e.g. line 461), I would encourage to use more appropriate formal language.

Author Response

Detailed comments:

Line 20-22: “the results for the tests carried out were, for a consumption of 100 kg/h, a lower heating value of 6.4 MJ/Nm3 …” This sounds like these were the conditions for the measurement, while this is actually part of the results already. Would suggest to split this and write something like: “A 50:50 mixture of almond husks and shells was found to have a lower heating value of …”

The reviewer's point is valid. The suggested modification to the abstract has been incorporated, and the text has been revised as per the suggestions provided.

Line 22: “Based on all the assumptions, a 100 kg/h (150 kWh) installation was proposed, …” Even after reading the whole article I did not understand why these parameters were chosen.

Regarding the question about the parameters chosen for the 100 kg/h (150 kWh) installation proposed in Line 22, we apologize for any confusion caused. We understand the importance of providing clear and comprehensive explanations for the rationale behind our choices. In the revised version of the article, we dedicate a specific section to elucidate the considerations that influenced these parameters.

Line 148-167

Line 25: “net present value of k€254,6, an …” Check comma.

Abstract in general: Information is missing on what the assumed end product of gasification is: The syngas that would be sold, biochar or electricity?

In the revised abstract, we have incorporated specific information clarifying that the primary output of the gasification process under consideration is syngas. This syngas has multifaceted applications, encompassing potential utilization for the generation of electricity.

Line 19.

Line 47: “One of the major benefits of biomass is its versatility in terms of application, as it can be used in combustion facilities of all sizes, …” This is no advantage of biomass compared to fossil fuels (especially oil & gas). They are actually much easier to scale and use for different purposes, such as in a gas cooker or huge gas power plants.

We understand the Reviewer perspective on the advantage of versatility of two fossil fuels, such as oil and gas, in terms of scalability and application in various situations, from the use of gas stoves to the operation of large gas-fired power plants. In fact, fossil fuels have been widely used for their ability to meet different demands on different scales.

The statement was corrected by analyzing some advantages of biomass, in which some specific points were addressed that differentiate two fossil fuels, despite the differences in versatility. Line 49-53.

Line 52: “biomass also offers environmental benefits, as its combustion generally leads to lower emissions of greenhouse gases and air pollutants, …“ Please be more precise. Combusting residual and annually produced biomass is generally carbon neutral. However, it does not necessarily lead to lower air pollutants (e.g. wood stove vs gas stove).

The observation is indeed accurate. Biomass combustion is often hailed for its potential environmental advantages, particularly in terms of lower greenhouse gas emissions when considering the carbon-neutral nature of combusting residual and annually produced biomass. This attribute arises from the fact that the carbon dioxide released during biomass combustion is offset by the carbon dioxide absorbed during the biomass growth phase. The manuscript was corrected.

Line 55-62

Line 80: “For this reason, the study validates …” Which study are you referring to? Yours?

The Reviewer is right, is our study. The text seems ambiguous, not knowing if it refers to literature or to our study. The statement was altered.

Line 139-143.

Line 89: I find this switch from the specific focus on almond shells and gasification (lines 65-88) to this very general section on electricity access very strange. In my opinion, the introduction should start general, and then become more specific. This jump between specific and general is not really nice to read. I would suggest to delete the paragraph (lines 89-103) and move the following paragraphs (from 104 onwards) more to the beginning of the introduction.

The reviewer is correct in addressing both the broader observations in the introduction and the more specific ones. The manuscript has been amended as suggested.

Line 63-123

Line 99: “By adopting such off-grid solutions …” Which off-grid solutions? And how does this fit together: Here you are talking about “off-grid solutions”, later (line 122) you refer to feed-in tarrifs, which obviously require connection to the grid.

Regarding the phrase on line 99, we recognize the ambiguity present in the mention of " By adopting such off-grid solutions …". This imprecision makes it difficult to fully understand the solutions that we are addressing in this context. To clarify this question, we revised the text and made an alteration to return to a more explicit reference. The phrase now is: "To adopt these decentralized energy solutions, including distributed generation and isolated systems..." We believe that this review will provide a clearer understanding of the solutions we are discussing.

Line 82-83 and 172-173

Line 145: The method section should start with an overview over your approach. Currently one can only guess what you may have done at this point.

In response to Reviewer suggestion, we have revised the method section to include a comprehensive and clear overview of the approach taken for the study.

Line 183-192

Line 154: This is an example where the text is much too long without providing any meaningful info. I would delete the subheadings 2.2.1 and 2.2.2 and just mention it under 2.2. Instead of the whole paragraph 2.2.1 you could just write something like “Nitrogen (N), carbon (C), hydrogen (H), sulfur (S), and oxygen (O) contents were determined with a Thermo Fisher Scientific Flash 2000 CHNS-O elemental analyser.” I think it is important to mention what a CHNS-O is, but really no more information is needed.

We agree that providing a brief mention of what the CHNS-O elemental analyzer, thermogravimetric and high heat value analysis are sufficient to convey the necessary information. We value your feedback and strive to present our methodology in a clear and concise manner.

Line 174: “byproducts or fuels” What you you mean here? The husk shell mixture?

The Reviewer is right, the term "byproducts" refers to any secondary products or substances that are produced during the gasification process, such as tar, char, or other chemical compounds. We make the change as suggested.

Line: 199

Line 219: “This equipment is capable of identifying and quantifying different gaseous elements present in the samples.” I think you shouldn’t assume that the readers are stupid. They will probably know that a gas chromatograph is capable of measuring gases. Just delete this sentence.

We apologize if the sentence came across as unnecessary. The Reviewer point is well taken, and we appreciate the perspective. We removed the mentioned sentence to streamline the text and ensure that the information remains concise and relevant.

Line:245

Line 274: I am really missing the description of the methods for the economic analysis here.

We have written a short sub-chapter of our manuscript incorporating a detailed explanation of the methods applied to economic analysis, ensuring that readers gain a complete understanding of the analytical approach.

Line: 295-344.

Lines 276-284: This should not be part of the results section. The results section should really only show results. Please delete this, and instead pick out and describe noteworthy results. I for instance find the low moisture content of husks and shells, and the low ash content of the shells interesting.

We appreciate your input and acknowledge the importance of maintaining a clear distinction between the methodology and results sections. We remove the sentence.

Table 1: These results of the ultimate analysis are related to the fresh matter content right? This should be mentioned in the caption. “Uni.” Is this an abbreviation for “Unit” ?? If so, please don’t do that.

The manuscript has been corrected as suggested.

Line: 349-350

Lines 288-289: “gasification, it is an innovative and promising process “ This is the results section!! Please don’t discuss here. Move to discussion or delete.

We delete the sentence, as suggest.

Table 2: Again “Uni.”. What do the “Sample” columns stand for? The different target temperatures? Then please rename. In line 301 you write of 6 samples, where are these represented in the table? Are the three just a random selection from these six? What is the difference between T1 and T2??

Thank you for the valuable observations and questions. We apologize for any confusion caused by the terminology used. Allow us to clarify:

In the "Sample" columns, each entry corresponds to a distinct temperature condition used during the analysis, we change the title for “Temperatures”. The designations "T1" and "T2" signify specific temperature zones. "T1" represents the temperature near the oxidation zone, while "T2" represents the temperature within the reduction zone of the reactor. Is clarify in the manuscript.

Regarding the mention of 6 samples in Line 301, we recognize the need for greater clarity. The three samples shown in the table are a temperature average of the set of 6 samples. The text has been revised to accurately convey this distinction.

Line: 388-394

Lines 295-306: This is all methods, and should not appear in the results section. Instead, a textual description of the results is missing.

We agree with the Reviewer. In response, we have removed that section and added a comprehensive textual description of the results, providing a more detailed overview of the findings and their implications.

Line: 400-418

Line 308: “Theoretical Parameters” Why are these theoretical parameters? Are they not real?

We removed the title and went straight into the discussion about gasification.

Sections 4.1.1.and 4.1.2: I think these are the least interesting sections. I would shorten them significantly. Also again, please do not assume that the readers are stupid. E.g. “Carbon is the primary constituent of organic materials and plays a fundamental role in the heating value of the fuel.” There is really no need to mention such basic things.

We have already revised and shortened sections 4.1.1 and 4.1.2 in response to the Reviewer suggestion. Additionally, we have removed these sections.

Figures 4, 5, 6: These are clearly results, and should therefore be moved to the results section, while lines 359-378 where you speculate on the process should remain in discussions.

We have addressed the Reviewer feedback by removing Figure 6 and relocating Figures 3, 4, and 5 to the results section as suggested. Additionally, we've ensured that lines 423-449 remain within the discussions section.

Figure 6: I think having three different scales / units in one graph is not good style.

We agree with the suggestion and have decided to remove the figure.

Lines 387-408: This leaves me confused. Which of the two parameters CCE or CGE is now more important to assess the performance? “The CCE (Carbon Conversion Efficiency) is a metric of utmost relevance …”?? Do you want to say that the CCE is the one that we should pay more attention to?

Certainly, I apologize for any confusion. The sentence in question should have conveyed that the "CGE (Carbon Gasification Efficiency)" is more relevant and significant than the "CCE (Carbon Conversion Efficiency)" when assessing performance. The CGE provides a clearer understanding of how effectively carbon is converted to gas, which is a key aspect in these assessments. The sentence has been rephrased.

Line: 472

I was wondering whether there is any difference between the different temperatures in terms of energy input. Do you need to preheat the gasifier somehow? Is there a difference in the energy input required? Are the results presented here for the whole runtime or only for after it has reached the target temperature? How would this affect results, if the gasifier was run 24/7 or for instance started each day in the morning?

There is no external heating source required once the gasification process has been initiated. An external ignition source is used to start the process, which typically takes around 10 to 15 minutes. Regarding the differences between temperatures, it's true that achieving higher temperatures does require a greater amount of oxidizing agent. However, the expectation is that with continuous operation (24/7), this requirement would stabilize and become quite similar for all temperatures.

The presented results encompass the entire runtime. If the gasifier were run continuously, the energy input requirements should eventually stabilize, minimizing the differences between the start-up phase (near 1h30m) and steady-state operation. This approach could potentially lead to more consistent and predictable outcomes, as opposed to variations that might be observed during intermittent starts.

Line 414: “4.3. Tcno-economic Analysis” Tcno??? Was Techno to long? Actually, what is coming is only the economic analysis anyhow.

Apologies for the confusion. The Reviewer is right, "Economic Analysis" would indeed be more appropriate. We've made the adjustment and changed the title to "4.3. Economic Analysis."

Section 4.2.1: As mentioned before, the economic assessment urgently needs its own section under “Methods”. The description here is too short and lacks important information.

We acknowledge the importance of providing comprehensive and detailed information in this section. In response to your feedback, we have included a dedicated section under "Methods" that focuses on the economic assessment.

Line: 294-344

Line 419: Here you mention a capacity of 150 kWe. I assume that the small ‘e’ stands for electric. You have not mentioned before how you would get electricity from the syngas. What were your assumtions for this? Which conversion efficiency did you assume? Did you conduct tests to figure this out? Would there maybe be any use for the residual heat from the syngas combustion? Also, you seem to only consider the use of the syngas, but are there any meaningful quantities of char that could be used? This should be mentioned in the results section as well.

In response to Reviewer inquiry about the capacity of 150 kWe, are indeed correct that the lowercase 'e' denotes electric. We apologize for any lack of clarity in explaining the methodology for obtaining electricity from syngas. We have now addressed this in the manuscript, specifically in lines 480 to 505, where we describe the process in detail. We have outlined our assumptions, including the conversion efficiency, and clarified the basis for our calculations.

Line: 485-502

Line 425: “approximately 720 tons per year will be required to obtain 150 kWe …” Is this a realistic size for a plant? For instance how many tons of husks and shells are collected per ha of almond farm, and what is the average farm size? Would this 150 kW plant fit for one farm, or for several together??

We appreciate the Reviewer insightful question regarding the scale of our plant's capacity. The issue that has raised is indeed pertinent, and we value the opportunity to clarify this aspect. In response to Reviewer query, we have adjusted our calculations and revised the electrical power to 100 kW electric, acknowledging previous miscalculations and extending our apologies for any confusion caused.

Considering the adjusted capacity, the concerns regarding the feasibility of this scale for an almond processing plant are valid. Our analysis did not account for the larger context of almond cultivation and the potential supply of raw materials. For instance, an almond farm typically yields around 2 tons of almond with husks per hectare, and assuming at least half the weight comprises husks, this equates to about 1 ton per hectare (reference: 10.17660/ActaHortic.2002.591.27). As exemplified by the 2018 investment by Route One Investment Company in Portugal, which encompassed approximately 4,000 hectares of almond cultivation, it is reasonable to anticipate an adequate supply of raw material to meet the energy demands of such processing plants.

Furthermore, the revised 100 kW electrical power capacity would indeed prove substantial for individual farms and possibly even collective operations. The excess energy generated could be channelled back into the grid or utilized for local consumption, thereby fostering sustainable energy practices, and addressing broader energy requirements, including almond processing campaigns.

Line: 514-528

Line 426: If you assume 7200 hours that means the plant would be running for the whole year, while the harvest happens in a short period of time. What are your assumptions on storing the husks and shells? Would they rot if they are just piled up? What is the current usage of these residues?

Given the pronounced seasonality inherent to this processing endeavor, prudent measures must be taken to ensure the preservation of the biomass feedstock. Accordingly, the storage of these husks and shells necessitates careful curation within a well-ventilated environment, marked by minimal humidity. This strategic approach is paramount in preserving the fundamental characteristics and energy content of the biomass.

In response to the Reviewer inquiries concerning the fate of these agricultural residues, we acknowledge their multifaceted utility within current practices. Traditionally, almond husks have served as a noteworthy energy source, predominantly as fuel for industrial furnaces, catering to sectors encompassing ceramics, greenhouses, and poultry farming. This established utilization speaks to the viability of these residues as a renewable energy resource, underscoring the potential synergy between the agricultural and industrial domains.

Line 520-528

Table 3: For the capital costs you assume 1300 € / kWh. Where is this data coming from? Are you using the regression from Bridgwater et al. 2002? As far as I can see this study only included gasifiers bigger then 1 MW, and costing more than 1 million €. I think you cannot just use this data. Also, the reference is more than 20 years old, and the values probably very outdated. Just from a brief search I could find several other potential sources, which partly use significantly higher values:

Colantoni et al. 2021 (https://doi.org/10.1016/j.egyr.2021.03.028) , referencing Sentis et al. 2016, consider total investment costs of 300.000 € for a 100 kWth gasifier.

Safarian et al. 2020 (https://doi.org/10.4236/jpee.2020.86001) considered 105.500 €, just for the hardware plus additional costs for installation.

Codoso et al. 2020, which are also cited by the authors, assume capital costs of 1760 € /kW for a 100 kW plant.

Allesina and Pedrazzi 2021 (https://doi.org/10.3390/en14206711) Provide an overview on companies and elaborate on maximal feasible  investment costs.

The Reviewer is right in questioning the origin of the initial assumption of 1300 €/kWh for capital costs. The observation regarding the Bridgwater et al. 2002 study is accurate; it predominantly includes gasifiers exceeding 1 MW in size and with costs surpassing 1 million €, rendering its applicability to our study questionable.

We have taken the feedback to heart and acted upon it. In line with the suggestions, we've revised our capital cost assumption to 1800 €/kWh, which aligns more closely with potential contemporary values. It's noteworthy that we encountered alternate sources in our investigation, some of which indeed cite substantially higher values. It's important to note that the field of gasification technology is evolving, with varying cost structures across different manufacturers. For instance, companies like Ankur Scientific and All Power Labs offer gaseifiers at considerably lower costs than the 1800 €/kW benchmark.

Since you define maintenance and operation costs relative to the capital costs, these assumptions on investment costs determine the whole result of the economic assessment. In such conditions, I would suggest that you do a sensitivity analysis based on a thorough literature research on potential capital costs.

We would like to assure you that we have indeed conducted a sensitivity analysis considering various economic factors, including maintenance and operation costs, based on our specified investment costs. We understand that investment costs play a pivotal role in shaping the overall economic evaluation, and we have strived to capture this relationship in our assessment.

Line: 611-657

Table 3 also mentions an “Electric kW price (€/kWh)”. What is this used for? Is it assumed that the plant consumes electricity from the grid? As far as I can see this is not mentioned anywhere.

we understand the Reviewer concern regarding the assumption of plant electricity consumption from the grid, which may not have been explicitly stated. We want to assure that this aspect has been addressed in the manuscript, where we discuss the context of the assumed electricity source and its implications on the economic assessment.

Line: 531-542

Line 453: You write that the results are for 800°C. Why was this temperature taken? Is this the best temperature based on the previous results? If so, I would like to understand why.

The decision to utilize 800°C was based on a comprehensive evaluation of preceding findings and their implications, as delineated in the manuscript.

Line: 488-494

Figure 7: You are using inconsistent abbreviations of Payback Period in the text and figure. Sometimes PB, sometimes PBP.

We have addressed the inconsistency in the abbreviations of the Payback Period (PBP) as highlighted in Figure 7. We have made the necessary corrections to ensure uniformity throughout the text and figure, utilizing "PBP" consistently.

Lines 461-464: “Thus, Figure 7 personifies the joint effort between technical robustness and economic excellence, shedding light on the path for informed and well-founded decision-making. In this scenario of innovation and relentless pursuit of sustainable ventures …” This is not appropriate language for a scientific publication.

In response to Reviewer feedback, we have removed the mentioned passage from the manuscript. We understand the importance of maintaining a formal and objective tone throughout the publication to ensure its scientific integrity.

The discussion lacks a comparison to the multitude of other studies on biomass gasification. There are dozens of other studies that have addressed similar questions. E.g. Colantoni et al. 2021 have considered hazelnut shells. Do they come to similar results and conclusions? If not, why?

We have extended our discussion to include a comprehensive comparison with a 3 of other studies on biomass gasification. We have examined various works that have explored similar questions and have included a specific reference to the study by Colantoni et al. (2021).

Line: 582-597

I think the conclusions section should really focus on the conclusions, and not reiterate the results.

Lines 541-554: The final document should probably not contain the instructions for filling out this section.

These instructions were not intended to be included in the final version of the document, and we have subsequently removed them to ensure the clarity and professionalism of the manuscript.

Reviewer 2 Report

- Please fix the typo here: 4.3. Tcno-economic Analysis  

- Conclusion should not contain any citation, move it to the discussions section. (for example, the mentioning about the Horizon Europe Research and Inno- vation Programme never appeared in the main body. But it suddenly appeared in the conclusion.). 

- please try to re-present table 2 in a graph-table integrated style so that readers can quickly see the status of each parameter in a visual manner but at the same time can refer to their exact values. 

Author Response

Comments and Suggestions for Authors

- Please fix the typo here: 4.3. Tcno-economic Analysis  

Thank you for pointing out the typo in Section 4.3's title. We apologize for the oversight. We have rectified the error by updating the title to "4.3. Economic Analysis" to ensure accuracy and consistency.

- Conclusion should not contain any citation, move it to the discussions section. (for example, the mentioning about the Horizon Europe Research and Inno- vation Programme never appeared in the main body. But it suddenly appeared in the conclusion.). 

As suggested, we have removed the citation from the conclusion. This change ensures that the conclusion remains a concise summary of the study's key findings and insights without introducing new information that has not been thoroughly discussed in the main body of the manuscript.

- please try to re-present table 2 in a graph-table integrated style so that readers can quickly see the status of each parameter in a visual manner but at the same time can refer to their exact values. 

 We acknowledge the reviewer's feedback regarding the visibility of table 2. In response, we have integrated the graphical representation of the table within the results chapter, adjacent to the tabular data. Our aim is to ensure the manuscript's visual clarity and presentation quality.